# Dietary Changes among Adults in The Netherlands in the Period 2007–2010 and 2012–2016. Results from Two Cross-Sectional National Food Consumption Surveys

**DOI:** 10.3390/nu13051520

**Published:** 2021-04-30

**Authors:** Ceciel S. Dinnissen, Marga C. Ocké, Elly J. M. Buurma-Rethans, Caroline T. M. van Rossum

**Affiliations:** 1National Institute for Public Health and the Environment (RIVM), 3721 BA Bilthoven, The Netherlands; marga.ocke@rivm.nl (M.C.O.); ellybuurma@planet.nl (E.J.M.B.-R.); caroline.van.rossum@rivm.nl (C.T.M.v.R.); 2Division of Human Nutrition and Health, Wageningen University and Research, 6708 WE Wageningen, The Netherlands

**Keywords:** dietary change, food consumption, food monitoring, educational level

## Abstract

Insight into dietary trends is important for the development and evaluation of dietary policies. The aim of this study is to describe changes in dietary intakes of Dutch adults and to evaluate these changes by age, gender, and education. In 2007–2010 and 2012–2016, two national food consumption surveys were conducted including 2106 and 1540 adults, respectively. Data collection included two non-consecutive 24 h dietary recalls. Mean habitual intakes of foods and nutrients relevant for a healthy diet of both surveys were estimated. Between the two periods the mean consumption of red or processed meat, dairy, sodium and alcohol and the ratio of whole-grain to cereal products decreased by 4–30% and the consumption of fibre and unsaturated fatty acids increased by about 3% and 6%, respectively. For most food groups, changes in consumption were comparable for both sexes and in all age groups. A healthier consumption pattern and several favorable changes were observed among higher-educated people. Most, but not all, changes in food consumption are favorable from a public health point of view. However, there is still a large potential for further improvements. A healthier consumption pattern was observed in adults with a higher educational level which calls for attention to social disparities when developing dietary policies.

## 1. Introduction

Diabetes, cardiovascular diseases and some types of cancers are chronic diseases that are, for the most part, related to poor diet and obesity [1,2]. To help achieve an optimal diet and reduce preventable diet-related illnesses, there is a need to develop effective nutrition and food policies. Based on this knowledge, health councils stipulate dietary guidelines for a healthy diet for an entire population [3,4,5,6]. These guidelines are evidence-based and aimed to reduce the risk of chronic diseases for the general population.

National food consumption surveys provide the opportunity to get insight into the dietary behavior of a population [7]. These surveys can therefore be used to help formulate and evaluate dietary policies. Repeated dietary surveys can help to gain insight into the development of dietary intake over time. To be suitable for this purpose it is essential that the repeated food consumption surveys are standardized.

Alongside population diet quality, surveys can identify disparities between subgroups. Diet-related risk factors are not distributed equally across population subgroups leading to dietary- and, possibly subsequently, health inequalities. There are gender differences in eating behavior, women tend to eat healthier foods than men do [8]. Additionally, with increasing age, people tend to eat less and make different food choices [9,10]. Poorer health and unhealthier lifestyles are more prevalent in individuals with a lower socio-economic position (SEP) than that for individuals with a higher SEP [11,12,13]. SEP is usually assessed by determining education, occupation and income. Education is considered to be most related to dietary behavior [14]. In line with this, groups who are less educated and have a lower income appear to consume a less healthy diet [15]. Few studies have described time trends in food consumption for a large variety of food groups and nutrients and studied if the observed trends differed by socio-economic groups.

The aim of the current paper is to describe dietary changes in the period 2007–2010 versus 2012–2016 of Dutch adults overall and by age group, gender, and educational level with data of the Dutch national food consumption survey (DNFCS) from 2007–2010 and 2012–2016.

## 2. Materials and Methods

Food consumption data from participants aged 19–69 years in two DNFCSs were used for the present analyses. Data from the first survey was collected between March 2007 and April 2010 among 3819 6–69-year-olds including 2106 Dutch adults aged 19–69 years. Data from the second survey was collected between November 2012 and January 2017 amongst 4313 1–79-year-olds including 1540 adults aged 19–69 years [16]. Both surveys largely followed the EU Menu guidelines [17].

In both surveys, participants were recruited via a consumer panel. Exclusion criteria were involvement in a food consumption survey during the previous four years, pregnancy or lactation, being institutionalized or inadequate command of the Dutch language. Sampling was stratified by age, gender, region, degree of urbanization and educational level. A questionnaire was used to obtain information on relevant background and lifestyle factors including educational level. The dietary assessment was based on two telephone-administered 24 h dietary recalls on non-consecutive days with at least 2–6 weeks in between. Interviews were conducted by trained dietitians using the computer-based interview program EPIC–Soft, now called GloboDiet [18]. The recalls were distributed over all days of the week and seasons. 

Mean habitual intake of foods and nutrients, associated with foods groups mentioned in the Dutch dietary guidelines set by the Health Council of the Netherlands were selected to be presented [3]. These guidelines recommend following a more plant-based and less animal-based dietary pattern. Daily or weekly dietary guidelines were created for fruits, vegetables, brown bread, wholemeal bread or other wholegrain, unsalted nuts, legumes, dairy, (oily) fish and tea. Consumption of salt and that of red meat, particularly processed meat should be limited, and the consumption of sugar-containing beverages should be minimized. It is advised to drink no alcohol or no more than one glass daily. We looked at intake of sodium and the nutrient alcohol rather than salt intake and the consumption of alcoholic beverages, respectively. For three food groups, the guidelines advise a replacement of specific foods rather than a consumed amount. It is advised to replace refined cereal products with whole-grain products and to replace butter, hard margarines and cooking fats with soft margarines, liquid cooking fats and vegetable oils. For those foods the habitual consumption of the advised subgroup was calculated as a percentage of the total food group, by dividing the mean intake of the subgroup by the mean intake of the total food group. It is also advised to replace unfiltered coffee with filtered coffee, however, our data cannot differentiate between these two states and is therefore not calculated.

Energy and nutrient intakes per person per day were calculated using the Dutch Food Composition Databases (NEVO). The NEVO table of 2011 was used for the 2007–2010 data and NEVO-online 2016 was used for the 2012–2016 data [19,20]. Vegetable protein and animal protein intake to study if a shift towards a more plant-based eating pattern was seen. The other nutrients were mono- and disaccharides (because of the limitation of sugar-containing beverages), fiber (because of the increase in whole-grain, vegetable and fruit consumption), unsaturated fatty acids (because of the replacement of solid fats to liquid fats and oils), n-3 fish fatty acids intake (because of the guideline on (oily) fish consumption), sodium intake (because of the limitation of salt) calcium and vitamin D intake (because of the dairy consumption) and alcohol (because of the limitation of alcoholic beverages).

Habitual daily energy-, nutrient- and food group intake distribution for each survey was estimated using RStudio software (version 4.0.2, RStudio, PBC, Boston, MA, USA) and the package ‘SPADE’ (Statistical Program to Assess Dietary Exposure, RIVM, version 3.2.55) [21]. SPADE is an R package which includes modeling options to estimate habitual consumption based on two consumption days for foods or nutrients that are consumed daily and by almost all participants (all nutrients and for bread and cereals, dairy and total fat) or episodically (all other food groups). If consumed episodically, SPADE took information on never-consumers available from the general questionnaire into account for alcohol. The habitual intake distributions were modeled as a function of age using data of all survey participants, after which intake distribution estimates were requested for adults aged 19–69 years. 95% Confidence intervals of the estimated mean food group consumptions and nutrient intakes were obtained by bootstrapping [21]. When the means of the two surveys had non-overlapping confidence intervals, the intakes were considered statistically significantly different. Due to low consumption frequency of unsalted nuts and legumes, we could not calculate their habitual consumption and consequently could not study the change in the consumption of those foods. 

Data were analyzed for the total population and stratified by educational level, age and gender. Educational level was categorized into low (primary education, lower vocational education, advanced elementary education), middle (intermediate vocational education, higher secondary education) and high (higher vocational education and university). Age was classified into three groups (19–30, 31–50, and 51–69 years). 

SAS software, (Version 9.4, SAS Institute Inc., Cary, NC, USA) was used to calculate frequency distributions to describe the study population. Weighing factors were applied in all analyses to correct for deviances from the Dutch population in sociodemographic characteristics (gender, age groups, region, level of education, urbanization) and for day of the week and season at the time of each survey.

## 3. Results

In the 2012–2016 survey, a lower proportion of participants were categorized in the low educational level group and a higher proportion of participants were categorized in a high educational level group compared to the previous survey. The proportion of overweight or obese participants and participants who indicated drinking alcoholic beverages was slightly higher in 2012–2016 than in the 2007–2010 survey. A lower proportion of 31–50-year-olds was seen in the more recent survey. The proportion of the two other age classes was higher. Smoking status, compliance with the physical activity guideline and dietary supplement use was comparable in both surveys (Table 1). 

Between 2007–2010 and 2012–2016, the mean consumption of red- or processed meat and dairy decreased (Table 2). The consumption of whole-grain products in gram and as a percentage of cereals and cereal products also decreased. For most food groups, the direction of the changes in food consumption was similar for both men and women. However, the change in mean consumption of sugar-containing beverages (excluding coffee and tea) and the consumption of whole-grain products as a percentage of cereals and cereal products did differ by gender. The consumption of the sugar-containing beverages only significantly decreased in women; and the decrease in the proportion of whole-grain products was only significant in men, while in women the consumed amount of whole-grain products decreased but not the proportion of total cereals. In both surveys, women consumed more fruit and tea, and less red or processed meat, dairy and sugar-containing beverages compared to men. The gender differences in fruit, red or processed meat and dairy consumption decreased in 2012–2016 compared to those in 2007–2010, while for sugar-containing beverages and tea the gender differences increased.

The mean consumption of red or processed meat and dairy decreased in all age groups (Table 3). Only in the youngest age group, the mean consumption of fruit increased and sugar-containing beverages decreased. The consumption of whole-grain products in total and as a percentage of cereals and cereal products decreased in the two oldest age groups. The percentage of soft fats to total fats increased in the youngest age group but decreased in the oldest age group.

In both surveys, adults aged 51–69 years consumed more vegetables, fruit, fish, tea and a higher proportion of whole grain products and less sugar-containing beverages than the 19–30-year-olds. The difference in fruit and sugar-containing drinks consumption and proportion of whole-grain products between the youngest age group and the other groups decreased in 2012–2016 compared to 2007–2010. In 2007–2010 the 19–30-year-olds consumed the same amount of dairy compared to the 51–69-year-olds, in 2012–2016 the oldest group consumed more dairy than the youngest age group.

The decrease in dairy consumption was observed in all educational level groups (Table 4). Other changes were only observed in one or two educational level subgroups. Only in the group with the highest educational level, the mean consumption of vegetables increased. Only in the middle educational level group, the proportion of whole-grain products of total cereals decreased and the percentage of soft margarines, liquid cooking fats and vegetable oils to total fat increased between the two time periods. Additionally, the consumption of red or processed meat decreased in the low- and high-educated groups, but not in the middle group. 

In both surveys, people with a high educational level consumed more vegetables, fruit and tea and less red or processed meat and sugar-containing beverages than people with a lower educational level. The difference between the highest educational level group and the lower educational level group, seen in 2007–2010, increased in 2012–2016 for vegetables, fruit, tea and the proportion of whole grains but decreased for fish and the percentage of soft margarines, liquid cooking fats and vegetable oils to total fat. In 2007–2010, dairy consumption was highest in people with a high educational level. The larger decrease in consumption in the subgroup caused this group to become the lowest consumers of dairy in 2012–2016. 

For most nutrients, the changes in mean habitual intake between 2007–2010 and 2012–2016 were in the same direction for both sexes as for the total Dutch population (Table 5). The mean habitual intake of fiber and unsaturated fatty acids increased and intake of animal protein, mono- and disaccharides, sodium, calcium, vitamin D and alcohol decreased significantly in the total population and in both men and women. Some differences between men and women were observed. Energy intake decreased in women and stayed stable in men. Vegetable protein intake decreased slightly in women, whereas an increase in intake was seen in men. Furthermore, n-3 fish fatty acids intake increased in men and was not significantly increased in women.

In both surveys, only the intake of mono- and disaccharides and fibers was higher in women than in men. The gender differences for the intake of animal protein intake, unsaturated fatty acids, calcium and alcohol decreased in 2012–2016 compared to that of 2007–2010.

Divided by age group, the mean habitual intake changed in a similar direction for the majority of the nutrients (Table 6). The intake of fiber, energy percentage unsaturated fatty acid and n-3 fish fatty acids increased and the intake of animal protein, mono and disaccharides, sodium, calcium and vitamin D decreased in all age groups. 

Energy intake only decreased in the youngest age group whereas, vegetable protein intake increased only in the oldest age group. Furthermore, alcohol intake only decreased in the 31–50-year-olds and in the 51–69-year-olds.

In both surveys, adults aged 51–69 years intake of energy, vegetable protein, mono- and disaccharides, unsaturated fatty acids and sodium was lower and animal protein, fiber and n-3 fish fatty acids, calcium, vitamin D and alcohol was higher than that of 19–30-year-olds. The age difference declined slightly between the two periods for most of the nutrients. The difference remained the same only for unsaturated fatty acids and vitamin D and increased for calcium.

For all educational level groups, the mean habitual intake showed a similar direction of change for almost all nutrients (Table 7). The energy percentage of unsaturated fatty acid in the diet increased and the intakes of animal protein, mono- and disaccharides, sodium, calcium, vitamin D and alcohol decreased.

Changes in the intakes of energy, vegetable protein and fiber intake differed across educational levels. Energy intake increased in the group with a high educational level, while it decreased in the other two educational level groups. Vegetable protein intake decreased in the group with a middle educational level and increased in the group with a high educational level. Fiber intake increased only in the groups with a middle and high educational level.

In both surveys, people with a high educational level consumed more vegetable protein (and less animal protein), fiber, n-3 fish fatty acids, calcium, and alcohol than people with a low educational level. For fiber and protein intake the changes resulted in a larger gap between the group with a low and a high educational level. For some nutrients, the association with educational level changed. In the 2012–2016 survey, the high educational level group had a higher energy intake and lower contribution of mono-disaccharides to the energy intake than the lower educational level group, while it was comparable between these two groups in the 2007–2010 survey. For unsaturated fatty acids and vitamin D, the intakes were comparable in the most recent survey while in 2007–2010 the intakes were lower in the higher-educated groups than in the lower-educated groups.

## 4. Discussion

This study found several positive changes in health-related dietary intake between the periods 2007–2010 and 2012–2016 among Dutch adults. The consumption of fiber and unsaturated fatty acids increased by about 3% and 6%, respectively. The mean consumption of red or processed meat, animal protein, mono-and disaccharides, sodium and alcohol decreased by about 4–30%. Some less favorable changes were also observed; the intake of calcium and vitamin D decreased, potentially due to the decrease in dairy, as did the ratio of whole-grain products to cereals and cereal products. In both surveys, it was observed that people with a high educational level consumed more vegetables, fruit and tea and consumed less red or processed meat, and sugar-containing beverages than people with a lower educational level. However, for most food groups, changes in consumption over time were similar across the educational level groups, as well as for both sexes, and all age groups. Our findings represent the most comprehensive evaluation of the change in relevant dietary habits among Dutch adults in the previous decades.

The favorable decrease in consumption of red or processed meat is reflected in a decreased intake of animal protein. This is not an issue as total protein intake was still sufficient in the Dutch population [7]. The decrease in sodium intake is probably not related to a change in dietary habits but a result of the efforts by food manufacturers to reduce the salt content of their products to comply with the “National agreement to Improve Product Composition: Salt, Saturated Fat, Sugar (Calories)” [22]. The intake of n-3 fish fatty acid also developed favorably. However, its higher intake was not associated with an increase in fish consumption. This suggests that the types of fish consumed changed to kinds richer in n-3 fish fatty acids. Similar changes were found in England in their surveys of 2001 and 2009 [23,24]. Another positive development was the decrease in intake of mono- and disaccharides. As the total consumption of sugar-containing beverages kept stable in most subgroups, this decrease might be explained by the result of several other changes in the food pattern. For instance, changes in the sugar content of sugar-containing drinks or in consumption of some types of beverages (less fruit juices; data not shown) or in consumption of sweet snacks (data not shown), fruit and dairy. It would be interesting to monitor the contribution of sugar-containing drinks to the mono-disaccharides as the food industry made agreements for reformulation of these products.

Not all observed changes between the two surveys were beneficial from a health perspective. The consumption of dairy decreased and with that, the intake of vitamin D and calcium. Since the mean habitual vitamin D intake recommended by the Health Council is not met, a decrease in the intake of vitamin D is therefore unfavorable [25]. Because one of the principal functions of vitamin D is that it stimulates the absorption of calcium from foods [26], a combined decrease with calcium is unfavorable, even though the habitual intake of calcium in the Dutch population is quite high and should be monitored to prevent further decrease [3]. A second unfavorable change was the decrease in the consumption of whole-grain products. This was not reflected in a decreased intake of fiber per MJ of energy, partly because energy intake decreased and partly because fiber intake from other sources was larger. As in both surveys, the intake is far below the adequate intake, an increase in the consumption of whole-grain products is important for reducing the risk of chronic diseases.

Our observed changes in food group consumption of the total Dutch population differ from reported changes in the period 1987–1998. The decrease in the consumption of vegetables and fruit in that 10-year period has stopped, and the increase in tea consumption before 2000 was not significant anymore when comparing 2007–2010 versus 2012–2016. Because of differences in food classification, time trends in the consumption of other food groups could not be compared [12]. Additionally, for other European countries time trends in consumption were reported. Similar to our results, several countries found no change in vegetable [27,28,29,30], fruit [27,28] or fish [27,29] consumption and a decrease in (red- or processed) meat consumption [27,28] in the total population. However, contrary to the results of this study, most countries did not find a change in dairy consumption [28,29] or found a decrease in bread and cereal consumption [28,30]. Observed differences could be different trends in other countries but might also be explained by different time periods studied.

Our results show only minimal dietary disparities in changes by age and sex. However, comparing the food consumption by age, most of the differences are in favor of adults in the oldest age group. This group consumed more of the favorable food groups such as vegetables, fruit and tea and a higher proportion of whole grain products and they consumed less sugar-containing beverages. This difference was already apparent in 2007–2010 and continued in 2012–2016. Furthermore, compared to men, women had a more preferable diet since they consumed more fruit and tea, and less red or processed meat, and sugar-containing beverages in both surveys. However, they consumed less dairy, this gender difference decreased in 2012–2016 compared to that in 2007–2010, while for sugar-containing beverages and tea the gender differences increased. Similar but not exactly the same age and sex differences were observed in other Western European countries such as Germany [29], the UK [31] and France [32].

Dietary disparities by educational strata were also observed with a diet in favor of adults in the high educational level group. In both surveys, people with a high educational level consumed more vegetables, fruit and tea and less red or processed meat and sugar-containing beverages than people with a lower educational level. Between the two surveys, the mean consumption of vegetables significantly increased in people with a higher-educational level and the consumption of red or processed meat significantly decreased in both the lower- and the higher-educational level group. Consequently, the gap between people with a low educational level and a high educational level broadened in 2012–2016 for vegetables and decreased slightly for red or processed meat. Across Europe, a higher education level is associated with higher fruit and vegetable consumption [33,34,35]. In accordance, the Helsinki Health Study survey found that the difference in the consumption of fresh vegetables and fish, increased in favor of people with a higher education level, between 2000–2002 and 2007 [36]. Outside of Europe, disparities by income and educational level also widened for the purchases of fruits, vegetables and the percentage of calories from sugar between 2008 and 2018 [37].

Even though some of the dietary changes seen are a step in the right direction, the Dutch population still does not reach the recommended dietary guidelines set by the Health Council for any of the describe food groups [7]. The guidelines are developed based on scientific evidence and examine the relationship between diet and the risk of disease. Additional health gains can be achieved when the Dutch adopt a diet more in accordance with the Dutch Dietary Guidelines. This would lower the risk of stroke, coronary heart disease, diabetes, colorectal cancer and mortality from any cause [38,39]. Life expectancy of people with a lower educational level is 6 years lower and they live 15 years longer in poorer health than people with a higher educational level. Contributing factors could be a difference in lifestyle and food consumption. The ecological aspects of the Dutch dietary guidelines have also been considered by the Dutch Health Council. In general, a more plant-based and less animal-based diet is associated with a lower ecological footprint [40]. The observed declines in the consumption of meat and dairy are also expected to result in lower greenhouse gas emissions of the diet [13,41]. More research on this topic should be done to get more insight and identify potentially favorable changes for the ecological impact.

Altering food consumption in people is known to be challenging. This is even more the case in subgroups with a lower educational level since they tend to have a poorer attitude toward healthy eating [42]. People with a higher educational level usually have more knowledge about healthier food items compared to people with a lower educational level [43]. Increasing dietary information could therefore be helpful. However, other policies, such as economic interventions, such as taxes aimed at reducing the cost for healthy foods) and also more upstream policies focused on access to decent wages and social support might be even more effective [44,45]. Insight needs to be gained in what types of dietary policies are most effective in altering food consumption in people with lower educational levels. Changes in dietary policies can be made accordingly and help cater to the specific needs of this group [46].

Potential limitations of our study should be considered. We were not able to evaluate the dietary trends on all recommendations of the Dutch food-based dietary guidelines because the data did not allow this (filtered coffee), or because consumption levels were too low to estimate habitual intake for the Dutch population (e.g., legumes, unsalted nuts). The differences in consumption could potentially be caused by differences in participant characteristics between the two surveys. However, we assume that the observed changes are hardly affected by these changes in the Dutch population. In addition, our study also showed trends stratified by gender, age group, and educational level. As with any population measure, dietary information is subject to random and systematic error. At the group level, underestimation of energy intake is common when self-reported methods are used among adults [47,48]. The mean underreporting increased by 3% between the two surveys [7] and might explain part of observed decreases in intake. Most of our observed differences were however larger than 3%. Another potential limitation is that only two consecutive surveys are used. It would be of added value to repeat these analyses when the results of the 2019–2021 survey become available.

Our study also has several strengths. This is the first study to investigate the change in food consumption and nutrient intake in the 21st century in Dutch adults and different subgroups. Multiple food groups and corresponding nutrients were evaluated using similar validated methods across the two DNFCS, making valid comparisons possible. We used detailed and representative data for the Dutch population during the period of data collection. Both surveys had a large sample size. The characteristics of participants in the survey are consistent with the Dutch population demographics at the moment of the surveys. With the exception of the number of subjects with a very low educational level and those with non-Western migration background, they might not be as well represented in the DNFCS in the low educational level group [7]. In any survey, the most vulnerable groups are usually difficult to reach to participate [49,50].

## 5. Conclusions

Several changes in food consumption and nutrient intake occurred between 2007–2010 and 2012–2016 in Dutch adults. Various favorable changes were observed, such as an increase in fiber (g/MJ) and unsaturated fatty acids (en%) intake and a decrease in red- or processed meat consumption, in intake of animal protein, mono-and disaccharides (en%), sodium and alcohol intake. Some less favorable changes were also observed; the consumption of dairy, the type of cereals calcium and vitamin D decreased.

A healthier food consumption pattern was still more observed in adults with a higher educational level. Future assessment of trends in food consumption and nutrient intake -in the Dutch population is necessary to evaluate whether the observed changes will continue over the following years or whether they were temporary and to monitor if disparities between educational level groups will change over time.

## Figures and Tables

**Table 1 nutrients-13-01520-t001:** Characteristics of adults (19–69 years) participating in the Dutch National Food Consumption Surveys in 2007–2010 (*n* = 2106) and 2012–2016 (*n* = 1540).

Characteristics	2007–2010		2012–2016	
	*n*	Weighted %	*n*	Weighted %
**Total**	2106		1540	
Men	1055	50.2	770	50.0
Women	1051	49.8	770	50.0
**Educational level**				
Low ^1^	708	32.1	323	25.6
Middle ^2^	935	43.6	655	43.4
High ^3^	463	24.3	562	31.0
**Age**				
19–30 years	703	21.5	516	22.4
31–50 years	699	44.7	523	40.7
51–69 years	704	33.9	501	36.9
**BMI classes**				
Normal and underweight ^4^	1010	45.6	714	42.8
Overweight/obese ^5^	1095	54.4	826	57.2
**Current smoker**				
Yes	544	24.2	362	23.9
No	1521	75.8	1167	75.7
**Alcohol use**				
Yes	1456	69.5	1155	73.6
No	649	30.5	385	26.4
**Meeting guideline for physical activity**				
No	1577	74.6	1155	76.6
Yes ^6^	529	24.4	353	23.4
**Dietary supplement use**				
Yes	916	44.6	663	43.1
No	1189	55.4	877	56.9

^1^ primary education, lower vocational education, advanced elementary education; ^2^ intermediate vocational education, higher secondary education; ^3^ higher vocational education and university; ^4^ BMI ≤ 25 kg/m^2^; ^5^ BMI > 25 kg/m^2^, ^6^ Physical activity >5 day and >30 min.

**Table 2 nutrients-13-01520-t002:** Habitual mean ^¥^ (95%CI) food consumption (g/day) * for Dutch adults aged 19–69 years in 2007–2010 (*n* = 2106; DNFCS 2007–2010) and 2012–2016 (*n* = 1540; DNFCS 2007–2010) for the total population and by gender.

Food Group	Total		Men		Women	
	2007–2010	2012–2016	2007–2010	2012–2016	2007–2010	2012–2016
	*n* = 2106	*n* = 1540	*n* = 1055	*n* = 1051	*n* = 770	*n* = 770
Vegetables	137 (134–139)	140 (136–144)	138 (134–142)	142 (136–148)	135 (131–139)	138 (133–142)
Fruit	102 (98–106)	106 (102–110)	90 (85–96)	96 (90–102)	114 (109–119)	116 (110–122)
Red or processed meat	**93** **(91–95)**	**80** **(77–82) ^1^**	**112** **(109–116)**	**97** **(93–101) ^1^**	**74** **(71–77)**	**63** **(59–66) ^1^**
Dairy	**372** **(369–375)**	**333** **(330–335) ^1^**	**412** **(408–416)**	**364** **(361–367) ^1^**	**332** **(329–336)**	**301** **(298–304) ^1^**
Fish	17 (16–18)	17 (16–19)	19 (17–20)	18 (16–21)	15 (14–17)	16 (14–18)
Sugar-containing beverages	328 (317–340)	307 (294–321)	376 (357–395)	372 (351–392)	**281** **(268–294)**	**243** **(228–259) ^1^**
Tea	230 (219–241)	245 (231–260)	164 (151–177)	165 (148–183)	296 (278–314)	325 (302–348)
Cereals and cereal products	**208** **(207–209)**	**203** **(202–205) ^1^**	234 (232–236)	237 (235–239)	**182** **(181–183)**	**170** **(169–171) ^1^**
Whole-grain products	**101** **(99–103)**	**93** **(90–95) ^1^**	114 (110–118)	106 (102–110)	**88** **(86–91)**	**79** **(76–82) ^1^**
Whole-grain products. Cereals and cereal products (%)	**49** **(48–49)**	**46** **(45–46) ^1^**	**49** **(47–50)**	**45** **(43–46) ^1^**	49 (48–50)	46 (45–48)
Fats	**28** **(28–28)**	**23** **(23–23) ^1^**	**33** **(33–33)**	**27** **(27–27) ^1^**	**23** **(22–23)**	**18** **(18–18) ^1^**
Soft margarines, liquid cooking fats, and vegetable oils	**21** **(21–22)**	**18** **(17–18) ^1^**	**26** **(25–26)**	**22** **(21–23) ^1^**	**17** **(17–18)**	**14** **(13–15) ^1^**
Soft fats **/total fats (%)	77 (76–79)	79 (78–81)	78 (77–80)	80 (78–82)	76 (74–78)	77 (75–79)

^1^ Sig. difference in mean consumption (95%CI) between DNFCS 2007–2010 and DNFCS 2012–2016, in bold; * except when indicated with (%) for substitution guidelines; ** Soft margarines, liquid cooking fats and vegetable oils; ^¥^ weighted for socio-demographic characteristics, season and day of the week.

**Table 3 nutrients-13-01520-t003:** Habitual mean^¥^ (95%CI) food consumption (g/day) * for Dutch adults aged 19–69 years in 2007 to 2010 (*n* = 2106; DNFCS 2007–2010) and 2012 to 2016 (*n* = 1540; DNFCS 2007–2010) by age groups (19–30, 31–50, 51–69).

Food Group	19–30 Years		31–50 Years		51–69 Years	
	2007–2010	2012–2016	2007–2010	2012–2016	2007–2010	2012–2016
	*n* = 703	*n* = 516	*n* = 699	*n* = 523	*n* = 704	*n* = 501
Vegetables	117 (113–120)	121 (118–125)	135 (132–139)	139 (134–144)	151 (146–156)	152 (147–157)
Fruit	**80** **(76–84)**	**92** **(87–98) ^1^**	97 (92–102)	100 (95–106)	123 (117–129)	121 (115–127)
Red or processed meat	**90** **(87–94)**	**76** **(73–79) ^1^**	**92** **(89–95)**	**81** **(78–85) ^1^**	**96** **(92–99)**	**80** **(77–83) ^1^**
Dairy	**376** **(372–381)**	**327** **(323–331) ^1^**	**372** **(368–376)**	**330** **(327–333) ^1^**	**370** **(366–374)**	**339** **(336–341) ^1^**
Fish	13 (11–14)	13 (11–15)	17 (15–18)	17 (14–19)	20 (17–22)	21 (18–23)
Sugar-containing beverages	**545** **(527–563)**	**491** **(471–511) ^1^**	330 (313–346)	313 (295–330)	189 (178–200)	190 (178–203)
Tea	184 (170–198)	194 (178–209)	231 (214–247)	268 (246–289)	259 (243–276)	252 (232–273)
Cereals and cereal products	**230** **(227–232)**	**218** **(214–221) ^1^**	215 (213–217)	212 (209–215)	185 (184–187)	185 (183–188)
Whole-grain products	92 (89–95)	85 (81–89)	**103** **(100–106)**	**95** **(91–99) ^1^**	**105** **(101–108)**	**95** **(91–98) ^1^**
Whole-grain products/Cereals and cereal products (%)	40 (39–41)	39 (38–40)	**48** **(47–49)**	**45** **(44–46) ^1^**	**56** **(55–58)**	**51** **(50–52) ^1^**
Fats	**26** **(25–26)**	**20** **(20–21) ^1^**	**28** **(27–28)**	**23** **(22–23) ^1^**	**29** **(29–29)**	**24** **(24–25) ^1^**
Soft margarines, liquid cooking fats, and vegetable oils	**20** **(19–20)**	**17** **(17–18) ^1^**	**21** **(21–22)**	**18** **(17–19) ^1^**	**23** **(22–23)**	**18** **(18–19) ^1^**
Soft fats **/total fats (%)	**76** **(75–77)**	**84** **(83–85) ^1^**	77 (76–79)	81 (79–82)	**78** **(77–80)**	**75** **(74–76) ^1^**

^1^ Sig. difference in mean consumption (95%CI) between DNFCS 2007–2010 and DNFCS 2012–2016, in bold; * except when indicated with (%) for substitution guidelines; ** Soft margarines, liquid cooking fats and vegetable oils; ^¥^ weighted for socio-demographic characteristics, season and day of the week.

**Table 4 nutrients-13-01520-t004:** Habitual mean ^¥^ (95%CI) food consumption (g/day) * for Dutch adults aged 19–69 years in 2007–2010 (*n* = 2106; DNFCS 2007–2010) and 2012–2016 (*n* = 1540; DNFCS 2007–2010) by educational level (low, middle, high).

Food Group	Low		Middle		High	
	2007–2010	2012–2016	2007–2010	2012–2016	2007–2010	2012–2016
	*n* = 708	*n* = 323	*n* = 935	*n* =655	*n* =463	*n* =526
Vegetables	126 (121–131)	124 (117–130)	131 (126–135)	128 (124–133)	**144** **(137–150)**	**167** **(160–175) ^1^**
Fruit	95 (89–101)	96 (87–104)	102 (97–108)	98 (92–104)	118 (110–126)	133 (126–141)
Red or processed meat	**98** **(94–102)**	**86** **(81–92) ^1^**	93 (90–96)	89 (85–93)	**82** **(78–86)**	**72** **(67–76) ^1^**
Dairy	**361** **(356–366)**	**339** **(333–345) ^1^**	**370** **(367–374)**	**344** **(340–348) ^1^**	**381** **(377–386)**	**316** **(313–320) ^1^**
Fish	14 (12–17)	19 (15–23)	16 (14–18)	15 (13–17)	21 (18–24)	21 (18–24)
Sugar-containing beverages	332 (310–353)	320 (287–354)	357 (339–375)	354 (333–374)	280 (259–301)	258 (239–276)
Tea	206 (188–224)	212 (182–242)	235 (218–252)	217 (196–238)	264 (236–292)	312 (280–344)
Cereals and cereal products	**196** **(193–199)**	**190** **(187–193) ^1^**	**216** **(215–218)**	**202** **(200–204) ^1^**	210 (207–213)	213 (211–216)
Whole-grain products	92 (88–97)	82 (75–88)	**105** **(101–109)**	**88** **(84–93) ^1^**	105 (100–110)	102 (98–107)
Whole-grain products/Cereals and cereal products (%)	47 (46–49)	43 (40–46)	**49** **(47–50)**	**44** **(42–45) ^1^**	50 (49–52)	48 (46–49)
Fats	**29** **(28–29)**	**23** **(22–23) ^1^**	**28** **(28–29)**	**22** **(22–23) ^1^**	**25** **(25–25)**	**22** **(22–23) ^1^**
Soft margarines, liquid cooking fats and vegetable oils	**21** **(21–22)**	**17** **(16–18) ^1^**	**21** **(20–22)**	**18** **(17–18) ^1^**	**19** **(19–20)**	**17** **(16–18) ^1^**
Soft fats**/total fats (%)	75 (73–78)	77 (73–80)	**75** **(73–77)**	**79** **(78–81) ^1^**	77 (74–80)	77 (74–80)

^1^ Sig. difference in mean consumption (95% CI) between DNFCS 2007–2010 and DNFCS 2012–2016, in bold; * except when indicated with (%) for substitution guidelines; ** Soft margarines, liquid cooking fats and vegetable oils; ^¥^ weighted for socio-demographic characteristics, season and day of the week.

**Table 5 nutrients-13-01520-t005:** Habitual mean ^¥^ (95%CI) * nutrient intake per day from food sources only for Dutch adults aged 19–69 years in 2007–2010 (*n* = 2106; DNFCS 2007–2010) and 2012–2016 (*n* = 1540; DNFCS 2007–2010) and by gender.

Nutrient	Total		Men		Women	
	2007–2010	2012–2016	2007–2010	2012–2016	2007–2010	2012–2016
	*n* = 2106	*n* =1540	*n* = 1055	*n* = 1051	*n* = 770	*n* = 770
Energy (MJ)	**9.5** **(9.5–9.6)**	**9.4** **(9.3–9.4) ^1^**	10.9 (10.9–11)	10.9 (10.9–11.0)	**8.2** **(8.1–8.2)**	**7.8** **(7.8–7.9) ^1^**
Vegetable protein (g)	**31.7** **(31.6–31.8)**	**32.1** **(31.9–32.3) ^1^**	**35.8** **(35.6–36.0)**	**36.9** **(36.7–37.0) ^1^**	**27.7** **(27.6–27.8)**	**27.3** **(27.2–27.4) ^1^**
Animal protein (g)	**53.9** **(53.8–54.2)**	**51.0** **(50.8–51.3) ^1^**	**61.5** **(61.2–61.8)**	**58.1** **(57.8–58.5) ^1^**	**46.5** **(46.3–46.7)**	**43.9** **(43.7–44.2) ^1^**
Mono- and disaccharide (en%)	**20.1** **(20.0–20.1)**	**19.4** **(19.3–19.5) ^1^**	**18.9** **(18.8–19.0)**	**18.3** **(18.2–18.4) ^1^**	**21.2** **(21.1–21.3)**	**20.5** **(20.4–20.5) ^1^**
Fiber (g/MJ)	**2.25** **(2.25–2.26)**	**2.31** **(2.30–2.31) ^1^**	**2.13** **(2.12–2.13)**	**2.20** **(2.19–2.20) ^1^**	**2.38** **(2.37–2.39)**	**2.42** **(2.41–2.43) ^1^**
Unsaturated fatty acids (en%)	**18.4** **(18.3–18.4)**	**19.4** **(19.4–19.5) ^1^**	**18.8** **(18.7–18.8)**	**19.8** **(19.7–19.8) ^1^**	**18.0** **(17.9–18.0)**	**19.1** **(19.1–19.2) ^1^**
n–3 fish fatty acids (EPA and DHA) (mg)	**132** **(125–140)**	**162** **(152–171) ^1^**	**137** **(127–146)**	**176** **(162–189) ^1^**	128 (117–139)	148 (137–160)
Sodium ** (mg)	**2745** **(2734–2755)**	**2582** **(2570–2595) ^1^**	**3124** **(3107–3142)**	**2972** **(2957–2986) ^1^**	**2367** **(2358–2375)**	**2193** **(2182–2203) ^1^**
Calcium (mg)	**1062** **(1059–1066)**	**1007** **(1003–1011) ^1^**	**1146** **(1141–1150)**	**1081** **(1076–1086) ^1^**	**979** **(974–983)**	**933** **(928–938) ^1^**
Vitamin D (µg)	**3.5** **(3.5–3.6)**	**3.1** **(3.1–3.2) ^1^**	**4.0** **(4.0–4.1)**	**3.6** **(3.5–3.6) ^1^**	**3.1** **(3.0–3.1)**	**2.7** **(2.6–2.7) ^1^**
Alcohol (g)	**14.0** **(13.3–14.8)**	**10.7** **(10.0–11.4) ^1^**	**19.5** **(18.2–20.9)**	**15.7** **(14.5–16.9) ^1^**	**8.6** **(7.7–9.5)**	**5.6** **(4.9–6.3) ^1^**

^1^ Sig. difference between mean intake (95%CI), DNFCS 2007–2010 vs. DNFCS 2012–2016, in bold; * unless indicated with (en%) for percentage of energy; ** Added salt is not included; ^¥^ weighted for socio-demographic characteristics, season, and day of the week.

**Table 6 nutrients-13-01520-t006:** Habitual mean ^¥^ (95%CI) * nutrient intake per day from food sources only for Dutch adults aged 19–69 years in 2007–2010 (*n* = 2106; DNFCS 2007–2010) and 2012–2016 (*n* = 1540; DNFCS 2007–2010) by age groups.

Nutrient	19–30 Years		31–50 Years		51–69 Years	
	2007–2010	2012–2016	2007–2010	2012–2016	2007–2010	2012–2016
	*n* = 703	*n* = 516	*n* = 699	*n* = 523	*n* = 704	*n* = 501
Energy (MJ)	**10.1** **(10.0–10.2)**	**9.6** **(9.4–9.7) ^1^**	9.7 (9.6–9.8)	9.7 (9.5–9.8)	8.9 (8.9–9.0)	9.0 (8.9–9.1)
Vegetable protein (g)	32.9 (32.5–33.2)	32.6 (32.2–33.1)	32.6 (32.4–32.9)	33.2 (32.8–33.6)	**29.8** **(29.5–30.0)**	**30.6** **(30.2–30.9) ^1^**
Animal protein (g)	**50.8** **(50.2–51.4)**	**47.6** **(46.9–48.3) ^1^**	**53.9** **(53.5–54.4)**	**51.9** **(51.4–52.5) ^1^**	**56.0** **(55.5–56.5)**	**52.1** **(51.6–52.6) ^1^**
Mono- and disaccharides (en%)	**22.3** **(22.2–22.4)**	**21.5** **(21.4–21.7) ^1^**	**19.9** **(19.8–20.0)**	**18.9** **(18.8–19.0) ^1^**	**18.9** **(18.8–18.9)**	**18.6** **(18.5–18.7) ^1^**
Fiber (g/MJ)	**2.12** **(2.11–2.13)**	**2.20** **(2.19–2.21) ^1^**	**2.24** **(2.23–2.25)**	**2.29** **(2.28–2.30) ^1^**	**2.36** **(2.35–2.37)**	**2.39** **(2.38–2.40) ^1^**
Unsaturated fatty acids (en%)	**18.4** **(18.3–18.4)**	**19.4** **(19.3–19.4) ^1^**	**18.5** **(18.5–18.5)**	**19.7** **(19.6–19.7) ^1^**	**18.2** **(18.1–18.2)**	**19.2** **(19.2–19.3) ^1^**
n-3 fish fatty acids (EPA and DHA) (mg)	**110** **(103–116)**	**136** **(128–144) ^1^**	**129** **(122–136)**	**165** **(156–175) ^1^**	**151** **(143–159)**	**174** **(164–184) ^1^**
Sodium ** (mg)	**2872** **(2837–2907)**	**2598** **(2561–2635) ^1^**	**2821** **(2798–2844)**	**2676** **(2647–2706) ^1^**	**2562** **(2541–2583)**	**2469** **(2442–2495) ^1^**
Calcium (mg)	**1028** **(1018–1037)**	**942** **(932–952) ^1^**	**1068** **(1062–1075)**	**1023** **(1016–1029) ^1^**	**1076** **(1071–1081)**	**1029** **(1023–1034) ^1^**
Vitamin D (µg)	**3.2** **(3.1–3.2)**	**2.8** **(2.8–2.9) ^1^**	**3.5** **(3.5–3.6)**	**3.1** **(3.1–3.2) ^1^**	**3.8** **(3.8–3.9)**	**3.4** **(3.3–3.4) ^1^**
Alcohol (g)	9.1 (8.0–10.1)	8.5 (7.3–9.7)	**13.1** **(11.9–14.3)**	**9.9** **(8.9–11) ^1^**	**18.4** **(17.1–19.8)**	**12.9** **(11.7–14.2) ^1^**

^1^ Sig. difference between mean intake (95%CI), DNFCS 2007–2010 vs. DNFCS 2012–2016, in bold; * unless indicated with (en%) for percentage of energy; ** Added salt is not included; ^¥^ weighted for socio-demographic characteristics, season, and day of the week.

**Table 7 nutrients-13-01520-t007:** Habitual mean ^¥^ (95%CI) * nutrient intake per day from food sources only for Dutch adults aged 19–69 years in 2007–2010 (*n* = 2106; DNFCS 2007–2010) and 2012–2016 (*n* = 1540; DNFCS 2007–2010) by educational level.

Nutrient	Low		Middle		High	
	2007–2010	2012–2016	2007–2010	2012–2016	2007–2010	2012–2016
	*n* = 708	*n* = 323	*n* = 935	*n* =655	*n* =463	*n* =526
Energy (MJ)	**9.4** **(9.4–9.5)**	**9.1** **(9.0–9.2) ^1^**	**9.7** **(9.7–9.8)**	**9.4** **(9.3–9.5) ^1^**	**9.3** **(9.3–9.4)**	**9.5** **(9.4–9.6) ^1^**
Vegetable protein (g)	30.4 (30.2–30.6)	30.1 (29.8–30.4)	**32.3** **(32.1–32.4)**	**31.8** **(31.6–32.0) ^1^**	**32.2** **(31.9–32.5)**	**34.0** **(33.7–34.3) ^1^**
Animal protein (g)	**54.2** **(53.8–54.6)**	**53.0** **(52.6–53.5) ^1^**	**54.1** **(53.7–54.4)**	**51.8** **(51.3–52.3) ^1^**	**52.1** **(51.8–52.5)**	**50.5** **(50.1–50.9) ^1^**
Mono- and disaccharides (en%)	**20.0** **(19.9–20.2)**	**19.6** **(19.4–19.8) ^1^**	**20.4** **(20.3–20.5)**	**20.0** **(19.9–20.1) ^1^**	**20.1** **(20.0–20.2)**	**18.6** **(18.4–18.7) ^1^**
Fiber (g/MJ)	2.20 (2.19–2.22)	2.22 (2.20–2.23)	**2.22** **(2.22–2.23)**	**2.25** **(2.24–2.26) ^1^**	**2.37** **(2.36–2.38)**	**2.42** **(2.41–2.43) ^1^**
Unsaturated fatty acids (en%)	**18.5** **(18.5–18.5)**	**19.4** **(19.3–19.4) ^1^**	**18.3** **(18.3–18.3)**	**19.2** **(19.2–19.3) ^1^**	**17.9** **(17.8–17.9)**	**19.5** **(19.4–19.5) ^1^**
n–3 fish fatty acids (EPA and DHA) (mg)	132 (117–146)	156 (129–183)	129 (118–141)	142 (131–153)	170 (151–189)	187 (171–203)
Sodium ** (mg)	**2703** **(2683–2723)**	**2545** **(2523–2566) ^1^**	**2813** **(2797–2828)**	**2596** **(2576–2616) ^1^**	**2661** **(2639–2683)**	**2558** **(2535–2581) ^1^**
Calcium (mg)	**1032** **(1024–1039)**	**968** **(960–976) ^1^**	**1067** **(1061–1072)**	**1013** **(1006–1021) ^1^**	**1101** **(1094–1108)**	**1030** **(1025–1035) ^1^**
Vitamin D (µg)	**3.7** **(3.6–3.7)**	**3.2** **(3.1–3.3) ^1^**	**3.6** **(3.6–3.7)**	**3.1** **(3.1–3.2) ^1^**	**3.3** **(3.2–3.4)**	**3.0** **(3.0–3.1) ^1^**
Alcohol (g)	**12.8** **(11.4–14.2)**	**8.8** **(7.1–10.4) ^1^**	**14.5** **(13.3–15.7)**	**10.9** **(9.8–12) ^1^**	**16.0** **(14.4–17.5)**	**12.3** **(11.0–13.6) ^1^**

^1^ Sig. difference between mean intake (95%CI), DNFCS 2007–2010 vs. DNFCS 2012–2016, in bold; * unless indicated with (en%) for percentage of energy; ** Added salt is not included; ^¥^ weighted for socio-demographic characteristics, season, and day of the week.

## Data Availability

The data used in this study are available on request from https://www.rivm.nl/en/dutch-national-food-consumption-survey/data-on-request (accessed on 25 March 2021).

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
