# Peer review of "Dietary Changes among Adults in The Netherlands in the Period 2007–2010 and 2012–2016. Results from Two Cross-Sectional National Food Consumption Surveys"

_nutrients, 2021, doi:10.3390/nu13051520_

Round 1

Reviewer 1 Report

Dinnissen et al. reported dietary change in Dutch adults between two study periods. The detailed descriptions of dietary intake and insightful discussion are informative for readers. However, there are several concerns, which should be sufficiently addressed to improve the quality of the manuscript.

1) Language editing by a native speaker may improve the quality of this manuscript. Further, the use of indent and comma should be checked.

2) Title

The present title is confusing. Although the authors estimated dietary intake cross-sectionally in each of the two study periods, readers may consider that dietary changes during the two periods were compared by difference-in-difference design. Please revise the title.

3) Abstract

“in the Dutch population” in line 18-19 should be deleted because all of the participants in the present study are Dutch.

4) Introduction

In line 46, an abbreviation is not SES but SEP and “higher socio-economic position” should be presented using the abbreviation.

Material and Methods

5) Line 60

The age of "2,106 Dutch adults" should be presented. Because the description of the age and the final number of participants duplicates in this section and line 130, I suggest the authors delete one of them.

6) Lines 69-73 and 92-94

Is there any difference in the procedures of dietary assessment methods as well as the nutrient contents in food composition databases used between the two study periods? If so, the authors should explain the kind and degree of difference and discuss its influence on the present findings in the discussion section.

Results

7) Although the detailed description of the results is helpful, it may be possible to shorten the text without losing the crucial points. I recommend the authors describe the tendency and essential points of the present findings in this section. The description of the abstract may be revised according to this revision. Additionally, Tables 2-4, as well as Tables 5-7, may be combined into one table.

8) Line 131-137

This section should refer to Table 1. Further, if the authors compare the characteristics of participants between the two study periods, they should conduct statistical tests and show the method in the method section and footnote in Table 1.

9) Line 189-190

"the low and middle” educational level group?

10) Discussion

The discussion section is well written and insightful.

11) Line 364-380

This section should be located after line 407. I do not understand the statement in line 378 because the degree of misreporting between two surveys is not guaranteed to be the same. Please show the rationale or revise the statement. Moreover, because of the strong correlation between energy intake and intakes of foods and nutrients, the difference in dietary intake between the two study periods as well as participants' characteristics also reflects the difference in energy intake. The authors should discuss this issue in this section.

12) Conclusion

The conclusion is also long. Please shorten that.

Author Response

Language editing by a native speaker may improve the quality of this manuscript. Further, the use of indent and comma should be checked.

  • The use of ident and comma’s are checked in the entire abstract and adjusted where needed.

Title; The present title is confusing. Although the authors estimated dietary intake cross-sectionally in each of the two study periods, readers may consider that dietary changes during the two periods were compared by difference-in-difference design. Please revise the title.

  • The title is changed to: Dietary changes among adults in the Netherlands in the period 2007-2010 and 2012-2016. Results from two cross-sectional national food consumption surveys 

Abstract; “in the Dutch population” in line 18-19 should be deleted because all of the participants in the present study are Dutch.

  • The text ‘in the Dutch population’ in the abstract has been deleted.

Introduction; In line 46, an abbreviation is not SES but SEP and “higher socio-economic position” should be presented using the abbreviation.

  • Abbreviation corrected to SEP and “higher socio-economic position” is abbreviated to SEP.

Line 60. The age of "2,106 Dutch adults" should be presented. Because the description of the age and the final number of participants duplicates in this section and line 130, I suggest the authors delete one of them.

  • Added the age of the 2,106 adults in line 60 and deleted the description of age in the result section (line 129-130), so duplication is prevented.

Lines 69-73 and 92-94. Is there any difference in the procedures of dietary assessment methods as well as the nutrient contents in food composition databases used between the two study periods? If so, the authors should explain the kind and degree of difference and discuss its influence on the present findings in the discussion section.

  • The same software for 24-h dietary recalls was used in both surveys so data collection was similar. However, based on this question, we looked again in detail to the procedures of food classification in the two surveys. We noticed that there were differences in two aspects . Some mixed dishes (pancakes, soups, and salads) were considered as foods and placed in one food group in 2007-2010 whereas they were split in ingredients after which each ingredient was placed in its own foodgroup in 2012-2016. We have adjusted the procedure for the 2007-2010 data and recalculated all results. In addition, not all fats consumed were correctly assigned to the right subgroups. We were able to solve these issues. Based on these changes, various results on consumption of food group changed and tables 2-4 were updated as well as the text. The changes also had implications for some of the conclusions.
  • Furthermore, as mentioned in the discussion, the surveys represent as good as possible the foods eaten by the population during the period of the surveys. The most appropriate version of the Dutch food composition database was used for each survey; so these were two different versions. Although these databases represent the foods eaten during the surveys best, we cannot exclude the possibility that some artificial differences might be due to the different food composition databases used. However some of our main finding, such as the observed decrease in salt intake, is in accordance with known changes in salt levels in foods.

Although the detailed description of the results is helpful, it may be possible to shorten the text without losing the crucial points. I recommend the authors describe the tendency and essential points of the present findings in this section. The description of the abstract may be revised according to this revision. Additionally, Tables 2-4, as well as Tables 5-7, may be combined into one table.

  • We have shortened the text by removing the description of mean intakes (which also can be seen in the corresponding tables).
  • The abstract has not changed since the alteration in the result section did not influence the content of the abstract.
  • The combination of table 2-4 and 5-7, would cause the tables to become very large and I feel, unclear. It would result in a table with 19 rows (one row for foodgroup or nutrient and 3 tables with 6 rows each, for total population and gender, age groups, educational level).

Line 131-137. This section should refer to Table 1. Further, if the authors compare the characteristics of participants between the two study periods, they should conduct statistical tests and show the method in the method section and footnote in Table 1.

  • Reference to table 1 has been added.
  • We have not conducted a statistical test since this is not the main focus of our study.

Line 189-190. "the low and middle” educational level group?

  • Thank you for this comment, this was incorrect. We have corrected it to low and middle educational level group.

The discussion section is well written and insightful.

  • Thank you.

Line 364-380. This section should be located after line 407. I do not understand the statement in line 378 because the degree of misreporting between two surveys is not guaranteed to be the same. Please show the rationale or revise the statement. Moreover, because of the strong correlation between energy intake and intakes of foods and nutrients, the difference in dietary intake between the two study periods as well as participants' characteristics also reflects the difference in energy intake. The authors should discuss this issue in this section. 

  • We have moved the strengths and limitations section of the study to the last paragraph of the discussion. (This is not indicated in track changes for readability reasons.)
  • The misreporting has indeed increased by 3%in the 2012-2016 survey compared to the 2007-2010 survey, However we assume that this cannot explain all changes. We have added this in the discussion section.
  • added: ‘The mean underreporting increased with 3% between the two surveys [7], and might explain part of observed decreases in intake. Most of our observed differences were however larger than 3%.’

Conclusion. The conclusion is also long. Please shorten that.

  • Shortened the conclusion by altering the summary of the foods and nutrients that changed favourably or unfavourably.

Reviewer 2 Report

This is a very interesting descriptive research on changes in the intake of food groups and nutrients in two periods (2007-2010 and 2012-2016) in Dutch adults.  I have only minor comments.

Comments:

  • Do you have data on physical activity as another characteristic that could be interesting to take into account?
  • Please add in the footnotes in tables 2-7 that the mean intakes of foods and nutrients were weighted by season and weekday, as you explained in the material and methods.
  • Do you have data on coffee?
  • Do you have data on the different alcoholic beverages consumed? Are there differences between types of alcoholic beverages among subjects with different educational level?
  • Why do you express the fibre as “g/MJ” instead of “g/day”?
  • Do you have data on foreigners?
  • Line 330-333, please could you extend a little bit of the changes observed in older surveys in the Netherlands?

Minor comments:

  • Abstract line 13, there is a comma missing between age and gender in “changes by age gender, and education”
  • Abstract line 14, please add “, respectively” at the end of the sentence “… 1,540 adults.”
  • Abstract line 21, please add subjects after “…among higher educated”
  • Introduction line 31, please put illness in plural “illnesses,” and add a comma.
  • Introduction line 33, please delete the comma and add a full stop after the references [3-6]. Please review the punctuation marks throughout the entire manuscript.
  • Line 44, please put choice in plural “choices”.
  • Line 47, please change “assed” for “assessed”
  • Table 1 was not referenced in the text.

Author Response

Do you have data on physical activity as another characteristic that could be interesting to take into account?

  • Thank you for your suggestion we have added data on ‘Meeting guideline for physical activity’  in table 1, where the characteristics of the adults per survey are described.

Please add in the footnotes in tables 2-7 that the mean intakes of foods and nutrients were weighted by season and weekday, as you explained in the material and methods.

  • Added the following footnote in table 2-7:. ¥ weighted for socio-demographic characteristics, season, and day of the week.

Do you have data on coffee?

  • We have data on coffee, however we cannot differentiate between filtered or unfiltered coffee. Since it is advised to replace unfiltered coffee be filtered coffee by the Health Council of the Netherland, coffee is not calculated. (also see line 89-90) in the Materials and Methods section. We have now also added a sentence in the limitation section of the discussion to indicate that we could not evaluate all dietary guidelines

Do you have data on the different alcoholic beverages consumed? Are there differences between types of alcoholic beverages among subjects with different educational level?

  • We only calculated the intake of the organic compound alcohol (ethanol) and not the different alcoholic beverages consumed. The Dutch Dietary Guidelines do not differentiate between types of alcoholic beverages, therefore we did not calculate these results.

Why do you express the fibre as “g/MJ” instead of “g/day”?

  • In the current Dutch guideline for dietary fibre (ref GR2006) the unit is g/MJ. For this reason we also expressed intake in g/MJ.

Do you have data on foreigners?

  • We only have data for the Dutch population in our surveys. In the survey of 2007-2010, 80 out of the 2106 participants have a migration background and in the survey of 2012-2016, 129 out of the 1540 have a migration background. The group is too small and diverse for stratified reporting of their dietary intake. Therefore we did not report on this.

Line 330-333, please could you extend a little bit of the changes observed in older surveys in the Netherlands?

  • We added some more details on the comparison with older surveys in the Netherlands in the discussion section. ‘Our observed changes in food group consumption of the total Dutch population differ from reported changes in the period 1987-1998. The decrease in the consumption of vegetables and fruit in that 10 y period has stopped, and the increase in tea consumption before 2000 was not significant anymore when comparing 2007-2010 versus 2012-2016. Because of differences in food classification, time trends in the consumption of other food groups could not be compared [12]. ’

MINOR:

Abstract line 13, there is a comma missing between age and gender in “changes by age gender, and education”

  • Added the comma.

Abstract line 14, please add “, respectively” at the end of the sentence “… 1,540 adults.”

  • Added respectively

Abstract line 21, please add subjects after “…among higher educated”

  • Added participants

Introduction line 31, please put illness in plural “illnesses,” and add a comma.

  • Made plural and comma added.

Introduction line 33, please delete the comma and add a full stop after the references [3-6]. Please review the punctuation marks throughout the entire manuscript.

  • Deleted the comma and added a period. Checked the punctuation in the entire manuscript.

Line 44, please put choice in plural “choices”.

  • Changed to choices

Line 47, please change “assed” for “assessed”

  • Changed to assessed

Table 1 was not referenced in the text.

  • The reference was added by table 1
